# State-behaviour feedbacks between boldness and food intake shape escape responses in fish (*Gasterosteus aculeatus*)

Isaac Planas-Sitjà [1] ✉ & Christos. C. Ioannou [2]

Consistent differences in intrinsic state, amplified through state-dependent behaviour, could explain the ubiquity of animal personality variation. Boldness is often positively associated with a high metabolism and food intake. Even though a high food consumption is known to compromise oxygen-demanding activities, the influence of food intake on anti-predator escape responses has rarely been considered. By conducting experiments with three-spined sticklebacks (*Gasterosteus aculeatus*) in a setup with real-time tracking and a decoy heron predator, we show that bolder fish benefited from a higher food intake than shy fish, and reacted faster to a predator attack when food intake was the same before being attacked. However, a higher food intake slowed down the escape responses. These results shine light on how the fitness of shy and bold tactics could be balanced in the wild: the faster reaction of bold fish is impaired by their higher food consumption.

Escape responses are a fundamental mechanism of survival for prey across taxa and have therefore been studied extensively[1,2]. Escape responses are the result of complex sensorimotor control in which individuals respond with a sudden motion away from the threat or predator stimulus, thus moving in a manner that maximises their individual chances of survival[1]. Escape responses, such as the C-start (escape reflex response) in fish[3,4], have been shaped by predation pressure, resulting in unpredictable escape behaviour that is thought to increase prey survival by reducing the predator's ability to effectively anticipate the movement of their target.

In contrast to this unpredictability, individuals within populations of the same species can differ consistently from one another in their behaviour[5,6], often referred to as 'animal personality variation'. This source of variation is widespread in animal populations and can have significant fitness consequences and wide-ranging ecological and evolutionary implications[5,7,8]. In this context, boldness, i.e. the willingness of individuals to take risks, has become one of the most commonly studied axes of personality variation in animals[9]. Boldness is frequently measured as the time spent vulnerable to predators (e.g., away from a refuge)[10,11] or in how quickly other behaviours are resumed after a disturbance[12,13]. These behaviours are believed to be the mechanisms which result in bolder individuals being at greater risk (although see ref. 8). Despite this potential cost, variation in boldness can be maintained within populations as a result of growth-mortality trade-offs[7]. These trade-offs, shaped by predation pressure[14–16] and experience (e.g., individual tactics could be reinforced or modulated by repeated exposure to predators[6,17]), lead to a dynamic interplay between

personality, risk perception, and survival, that could influence individual escape behaviour.

Bolder individuals often benefit from a higher food intake[18], although at the expense of physiological and behavioural adaptations to meet their greater energetic needs[13]. It has been proposed that such individuals must possess some mechanism to decrease predation risk[8,19]. For instance, recent studies suggest that bold individuals may react faster to a predator threat than shy individuals due to their higher metabolisms[19,20]. Additionally, bold fish could learn how to avoid a predator faster merely because they encounter and interact with them more often[16,21,22]. These mechanisms could potentially explain why, in some species, bold individuals have higher survival rates compared to shyer individuals[8,23,24]. However, these findings are contrary to other studies suggesting that allocating investment in growth or competitive ability can come at the expense of cognitive investment[7,15], which can negatively affect anti-predator responses of bold individuals. Similarly, it has been suggested that bolder individuals may be less likely to respond, or respond more slowly, to a predatory threat because they allocate a greater proportion of their attention to foraging rather than detecting predators[25]. In addition to this attentional cost of foraging, once food is consumed, digestion can compromise other oxygen-demanding activities[26] such as anti-predator escape responses, thus potentially reducing the chances of escape. The trade-off between foraging and predation risk that is believed to generate variation in boldness within a population can thus be magnified via positive feedback from consuming food. Such 'state-behaviour feedbacks' have been argued to influence inter-individual variation[27]

[1]Animal Ecology, Department of Biological Sciences, Tokyo Metropolitan University, 1-1 Minami-Osawa, Hachioji, Japan. [2]School of Biological Sciences, University of Bristol, Life Sciences Building, 24 Tyndall Avenue, Bristol, United Kingdom. ✉e-mail: iplanass@pm.me

and could help explain the discrepancies mentioned above. In addition, their effect on escape responses has the potential to impact predator-prey interactions and hence larger scale community dynamics[28].

Fish, and three-spined sticklebacks (*Gasterosteus aculeatus*) in particular, have been used extensively as a model system for the study of inter-individual behavioural variation, including state-behaviour feedbacks[18,27,29,30]. These animals are ideal as an individual's state (e.g., hunger or information) can be controlled under laboratory conditions, and large numbers of wild-caught individuals can be kept under laboratory conditions. Three-spined sticklebacks show consistent heritable variation in their behaviour over generations[10,31], which may result from adaptive evolution in natural populations[32].

In this study, we investigate how boldness and food intake shape antipredator escape responses. For this purpose, we tested individual sticklebacks over several trials in a novel setup with precise control of a predator model by coupling real-time tracking of the fish with the automated predator triggering, which allowed a high degree of standardisation of the threat stimuli. We analysed the fish's consumption of food, probability to freeze, their latency to escape, and other kinematic measurements during the escape response (Fig. 1). With this approach we investigated repeatability over multiple days in the fish's boldness, motivation to feed, and escape responses, and aimed to determine how boldness and food consumption may influence the reaction to a threatening stimulus. We predicted that if bold fish react slower than shyer fish due to attention bias or limited cognitive capacities, food intake should play a minor role in escape responses. Alternatively, if bolder fish react faster than shy fish as an adaptation to decrease predation risk, we expected that the presence of food during a predator's attack may have a higher impairment on the escape response of bolder fish compared to shyer fish, particularly if bolder fish consumed more food before the attack.

## Results

Our results demonstrate that individual fish were consistent over the three days of testing in their latency to first leave the refuge ($R_c \pm$ SE = 0.37 ± 0.15, CI = [0.03, 0.62], $P = 0.03$), total distance travelled ($R_C$ = 0.3 ± 0.13, CI = [0.03, 0.55], $P = 0.008$), and the number of bloodworms eaten after the predator's attack ($R_C$ = 0.49 ± 0.18, CI = [0.15,0.84], $P = 0.001$), while the number of bloodworms eaten before the attack was close to the significance threshold ($R_C$ = 0.23 ± 0.16, CI = [0, 0.53], $P = 0.08$). Fish that had lower latencies to first leave the shelter also ate more bloodworms (before and after the attack) and tended to travel longer distances (Fig. 2). The latencies to reach the food patch did not differ over the three days of testing (ANOVA: $F_{2,58} = 0.031$, $P = 0.97$), suggesting that learning the location of the food patch did not have a noticeable influence on individual behaviour (Supplementary Fig. 1). Additionally, there was no indication that the three repeated trials and the encounter with the predator influenced the risk perception of the fish, as the latency to first leave the shelter (ANOVA: $F_{2,58} = 1.13$, $P = 0.33$) and the total distance travelled (ANOVA: $F_{2,58} = 0.382$, $P = 0.68$) did not differ over the repeated trials (Supplementary Fig. 1). In general, the total distance travelled over repeated trials had a higher degree of plasticity compared to the latency to leave the refuge or reach the food patch (Supplementary Fig. 1).

We conducted GLMMs to investigate whether multiple components influenced different response variables. We used the latency to first leave the refuge as a measure of boldness (tendency to take risk), and thus bold and shy fish are those with a low or high latency, respectively. The number of bloodworms eaten before, and after, the attack of the predator were used, respectively, as a measure of food intake before escape, and as a measure of their motivation to forage under the risk of predation after the attack.

When the predator model appeared, the probability to freeze or flee did not depend on boldness, body length, trial day or angle of the attack (GLMM1 (binomial); supplementary table 1). However, there was moderate evidence from the AICc model comparisons that the number of bloodworms eaten before the predator's attack improved the

likelihood of the models (adj. $R^2$ = 0.45, supplementary table 2). The probability of fleeing decreased as the number of bloodworms eaten increased (estimate ± SE = −0.67 ± 0.4, z = −1.69, $P = 0.06$; Fig. 3A).

The latency to escape (GLMM2 (Gaussian); adj. $R^2$ = 0.75; supplementary table 3), increased with the number of bloodworms consumed before the attack (estimate = 0.16 ± 0.056, $t = 2.91$, $P = 0.007$), and bold fish reacted faster than shy fish (estimate = 0.18 ± 0.065, $t = 2.71$, $P = 0.01$) (Fig. 3B). Fish showed a longer latency to escape when the attack came from behind the fish, that is with larger angles of the attack (estimate = 0.15 ± 0.065, $t = 2.3$, $P = 0.03$). We also found that fish reacted more slowly over the trial days, although this effect was weak (estimate = 0.098 ± 0.03, $t = 3$, $P = 0.005$; supplementary table 4). Fish body length and distance from predator had no effect on their latency to escape (supplementary table 3).

For fish that actively fled, the turn rate of the fish while they fled (GLMM3 (Gaussian); adj. $R^2$ = 0.79; supplementary table 5, 6) was faster at a further distance from the predator (estimate = 146.76 ± 71.87, $t = 2.042$, $P = 0.04$), and with a greater degree of bending of the body or lower curvature index (estimate = -400.55 ± 82.16, $t = -4.88$, $P < 0.001$). Consistent with this, the maximum curvature of the body as a response variable (GLMM4 (Gaussian); adj. $R^2$ = 0.65; supplementary table 7, 8) increased with the turn rate (estimate = -0.05 ± 0.01, $t = −5.18$, $P < 0.001$) and with the distance from the predator (estimate = 0.022 ± 0.01, $t = 2.48$, $P = 0.01$). Turn rate and curvature index were not affected by boldness, food intake, body length or trial day. For the initial speed of the escape response there was not strong evidence that any variable affected their initial speed (supplementary table 9).

After the attack, shyer fish resumed their activity later than bolder fish (GLMM6 (Poisson); estimate = 0.16 ± 0.02, z = 7.78, $P < 0.001$; adj. $R^2$ = 0.99; supplementary table 10, 11). In addition, the latency to resume activity also increased for fish with higher food intake before the attack (estimate = 0.2 ± 0.02, z = 10.88, $P < 0.001$; Fig. 3C), fish with shorter body length (estimate = -0.42 ± 0.07, z = −6.28, $P < 0.001$) and fish that swam longer distances during the escape (estimate = 0.59 ± 0.02, z = 30.26, $P < 0.001$). There was also an effect of day, with fish showing a tendency to increase their recovery time over the trial days (estimate = 0.03 ± 0.01, z = 4.94, $P = 0.06$). However, the distance from the refuge did not influence their recovery time.

Finally, we did not observe any significant repeatability of the escape behaviours measured as the maximum curvature ($R_C \pm$ SE = 0.3 ± 0.22, CI = [0, 0.714], $P = 0.13$) and escape angle ($R_C$ = 0.13 ± 0.19, CI = [0, 0.62], $P = 0.35$), as well as for the recovery time ($R_C$ = 0 ± 0.06, CI = [0, 0.2], $P = 0.5$). However, the fish did show high repeatability in their turn rate ($R_C$ = 0.64 ± 0.17; CI = [0.18, 0.86], $P < 0.001$).

## Discussion

In this study we investigated how boldness and food intake affect the escape of fish in response to an ambush predator model, a robot heron. Our results provide empirical evidence of a behavioural syndrome linking boldness (latency to leave a refuge) and motivation to feed (bloodworms consumed before or after the predator's attack); fish leaving the refuge earlier ate more than shyer fish. Boldness was only weakly correlated with the exploration tendency (total distance travelled), probably due to the high degree of plasticity in the total distance travelled. When the predator model attacked, the tendency to freeze rather than actively flee increased with the number of bloodworms eaten, but was unaffected by the fish's boldness. From those individuals that escaped (flight response), bolder fish reacted faster, and resumed activity sooner, than shy fish. However, fish consuming more food before the attack showed a slower escape response and resumed their activity later.

Within a single experimental setup, our study links risk-taking tendency to food consumption, and both to escape responses of an ecologically-realistic threat. The pace-of-life hypothesis posits that the bold-shy continuum arises from a trade-off between foraging gains and predation risk[5,7,33–35]. It is assumed that bolder individuals suffer from higher mortality

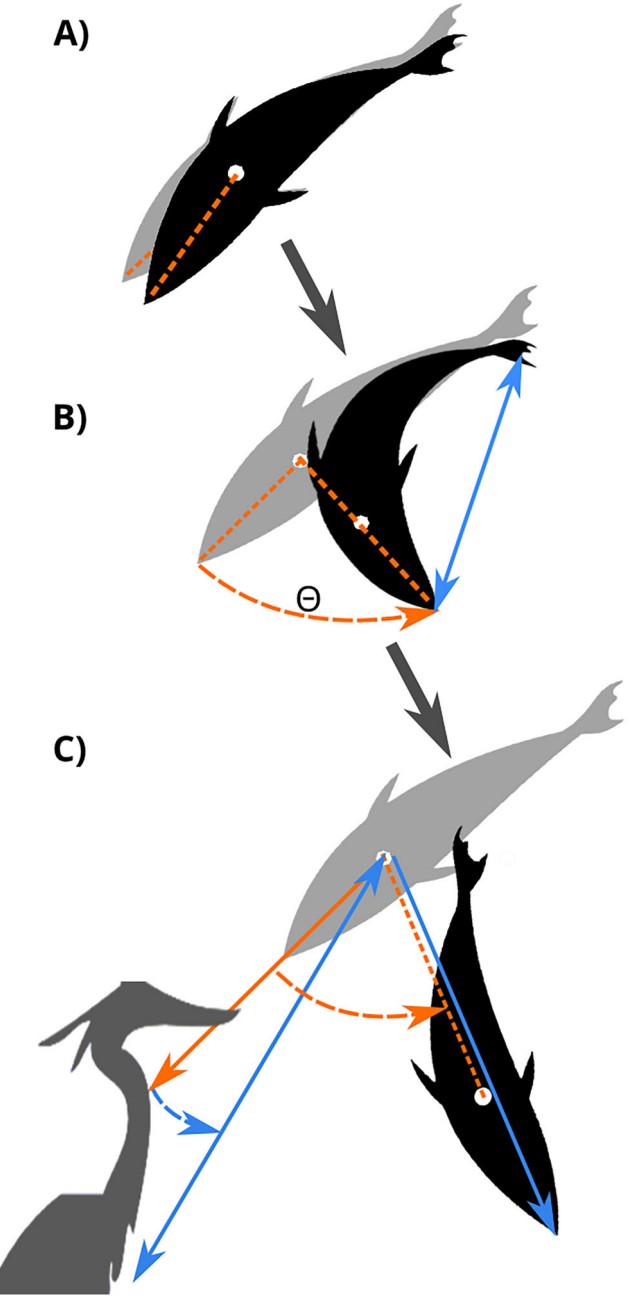

**Fig. 1 | Schematic of the escape response measures.** The grey fish silhouette represents the fish's position prior to starting the escape reaction, the black fish silhouette represents the fish's position during different stages of the escape response, and the white dot indicates the estimated centre of mass of the fish. **A** The escape latency was measured as the time that elapsed from the appearance of the predator (when the predator was first visible from the centre of the food patch) to the time that the fish first reacts. **B** The turn rate (Θ) was measured by dividing the angle achieved during the first unilateral bend of the reaction (orange dashed lines) by the duration of time to achieve that angle, with higher turning rate indicting greater agility in the response[71]. The curvature index was calculated as the minimum distance between the head and tail during the escape (blue double arrow line) divided by the fish's body length (thus an index of 1 indicates that fish did not bend). **C** The distance from the predator is measured as the distance between fish's position prior to the attack and the edge of the tank at the point where the predator is first visible (blue double arrow line). Initial speed was measured as the distance travelled (orange dashed line) during the first 42 ms of the reaction (equivalent to 10 frames, chosen for comparison to previous studies[71]). The escape direction was measured as the direction of the fish at 42 ms after the initial reaction (blue arrow). The angle of escape was derived from the escape direction relative to the orientation of the fish before the attack (orange dashed angle). The angle of the attack was defined as the angle formed by the orientation of the fish before the attack and the predator's location (blue dashed angle).

models predicting the escape responses suggests that bold and shy fish were equally impaired by food consumption. However, as bolder fish tended to eat more, the impairment of their escape responses will be greater.

Together, these results suggest that while bolder individuals may be better able to respond to a predatory threat and recover more quickly, this works antagonistically with their tendency to consume more food. Such state-behaviour feedbacks have been speculated to exist for some time[27] but, until now, have had relatively weak empirical support[18,40–42]. Most studies on state-behaviour feedbacks have focused on intrinsic states (e.g., metabolism, hormone levels), and while these relationships are important, they generally account for small proportion of the among-individual variability[19,42,43]. It has been suggested that the inclusion of factors different from intrinsic states may be important to explain the remaining variability[42]. In this context, environment can play a key role in determining the emergence of state-behaviour feedbacks and demographic processes, as some environments may be more selective than others[18,44–46]. For instance, environments with abundant and predictable food resources may benefit bolder, more active, and more aggressive fish. However, shyer, more passive, and less aggressive fish may obtain similar, or even higher, fitness gain as bolder individuals in environments with limited resources[47,48]. In addition, hunger state or the ability to locate food has been suggested to be as important as mortality risk in shaping reaction norms[46,49]. In the case of predator-prey interactions, a recent study on insects (*Athalia rosae*) provided evidence that starved larvae were more active and located food faster than non-starved larvae, although they were also attacked more frequently by a predator[50]. In black-headed chickadees, individuals that had higher feeding rates or resumed their activity faster were more easily recaptured[51], although the recapture rate did not vary as a function of exploration, aggressiveness or boldness. These results, together with our findings, suggest food intake is a powerful mechanism for state-behaviour feedbacks, and highlight the importance of integrating multiple dimensions of state when investigating the relationship between behaviour and fitness[28,49]. It should also be noted that seemingly high risk behaviour does not necessarily translate to realised mortality when risk-prone individuals possess mechanisms to decrease mortality. These mechanisms, whether cognitive, physiological or physical, could help bold individuals compensate for their increased exposure to predators during foraging. In other words, individuals may not differ only in their physical abilities to escape from predators, but also in their cognitive abilities. For instance, previous studies have shown that boldness in guppies was positively associated with their ability to learn an associative task[52,53]. While brain size and cognitive abilities may come at a cost of slower growth[7,54], bolder or more exploratory individuals can learn faster simply because they encounter

because they take more risk in order to obtain more food, even though they can have faster reactions due to their higher metabolisms[20,23,36,37]. In contrast to predictions from the pace-of-life hypothesis, a recent meta-analysis found that bolder individuals do not have shorter lives than shyer individuals in natural environments[8]. This suggests that individuals exhibiting a bold phenotype may possess mechanisms that allow them to reduce the costs associated with their risky behaviour. We found that bold fish reacted faster than shy fish, which supports the idea that bolder individuals can reduce the predation cost associated with foraging. However, our results also demonstrate that bolder individuals consume more food, and this hinders their response to a predatory threat; this is consistent with previous studies that show prey engaged in foraging are less likely to react, and tend to show shorter reaction distances than prey that are not feeding because of the attentional cost of foraging[38,39]. This effect could be further amplified over time as the energy spent on processing ingested food could compromise other oxygen-demanding activities such as escape responses[26]. The lack of importance of the interaction term between boldness and food intake in the

https://doi.org/10.1038/s42003-025-07669-w                                                                      **Article**

the risk (or cognitive task) quicker and/or more frequently than shyer individuals[6,15,16,52]. These mechanisms could thus explain, and amplify, differences in risk perception among individuals differing in boldness.

This study provides empirical evidence that bolder individuals have a higher motivation to feed, and that bolder individuals possess mechanisms (i.e., faster reaction to a threatening stimulus) that allow them to decrease the cost associated with risky behaviour. However, we show that food intake amplifies predation risk by slowing down the escape reaction. While being bold can have a positive effect on foraging, the tendency to eat more can have an indirect negative effect on mortality. These two antagonistic effects may help explain how predation can shape behavioural syndromes in nature.

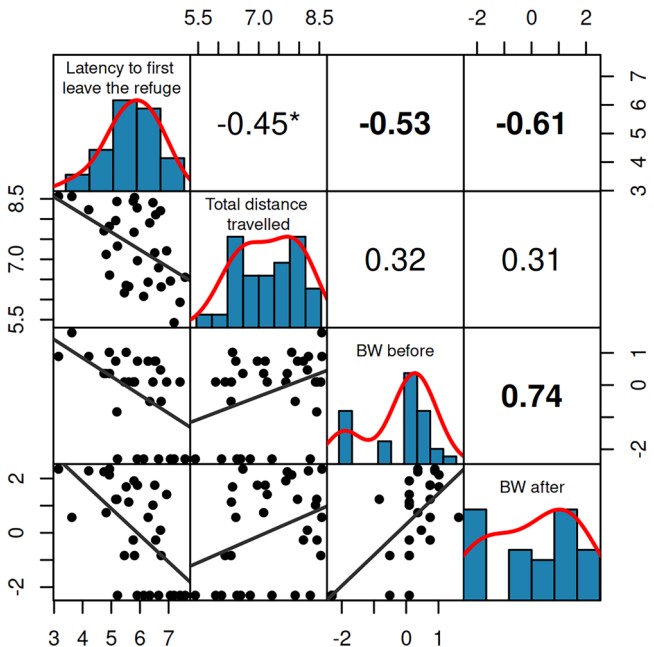

**Fig. 2 | Correlations among measurements: latency to first leave the refuge, total distance travelled, bloodworms eaten before the attack (BW before), and bloodworms eaten after the attack (BW after).** The numerical values (top-right) are Spearman's rank correlation coefficients of the corresponding scatter plots (bottom-left). Bold values indicate statistical significance after correcting for multiple comparisons (Bonferroni method), and asterisk indicates significant values before correction. All variables were log-transformed for clearer visualisation.

## Methods

### Experimental subjects

Fish ($36.7 \pm 5.5$ mm mean $\pm$ SD; $N = 42$) were caught from the River Cary, Somerset, UK (ST 469 303), a river of ~ 3 m width and less than 1.5 m depth (except some zones of > 2 m depth) with an overall simple and exposed habitat at the time of collection. The riparian vegetation coverage was low, and there were several small zones with aquatic vegetation that provide refuge to fish. Three investigators walked inside the river and caught fish with large hand nets approximately 50 cm in diameter; most fish were caught in the aquatic vegetation. While we could not assess the level of predation, the water had medium levels of turbidity, and the main predators observed in the area were pike trout, bass and piscivorous birds (e.g., herons, kingfishers). Fish were transported to the laboratory by car in a large foam box to keep temperature constant, and kept in $40 \times 70 \times 34$ cm (width × length × height) glass tanks after a two-week quarantine period. They were held in the laboratory for at least two months before experiments, and were fed flake food and defrosted bloodworms once per day, and only after testing on experimental trial days. Water temperature throughout was 14 °C, and lighting was on a 11:13 day:night cycle. Fish were not sexed because under these conditions fish were not in reproductive condition[55], and there is no relationship between sex and boldness before sexual maturation[56]. All fish were used in this study, and no fish died during the transport, quarantine or experimental period.

### Experimental setup

The setup was designed to study the escape behaviour of individual fish when facing a predator from above. The white acrylic glass experimental tank (1 m × 1 m; water depth = $7 \pm 1$ cm) was divided into two arenas with a white corrugated polypropylene plastic divider, thus creating two identical experimental arenas of 1 m × 0.5 m. The setup was surrounded by white curtains to avoid any disturbance. A refuge area of $25 \times 22 \times 12$ cm, containing plastic plants to provide cover, was placed at one end of each arena. A white corrugated polypropylene plastic door (23 cm width) controlled access to the main arena, and a stepper motor 28BYJ-48 (with a ULN2003 driver) activated with Arduino opened and closed the door at a gradually increasing speed to minimize disturbance on fish behaviour. The door remained open until the end of each trial. On the other side of the arena to the refuge, a 90 mm Petri dish, referred to as the food patch, was attached to the bottom of the arena at the centre of the short axis of the arena, and 24 cm from the end of the arena. The food patch contained $13 \pm 1$ small bloodworms for each trial. A plastic heron predator model (Pisces brand) was positioned behind the food patch before it was activated so that it could not

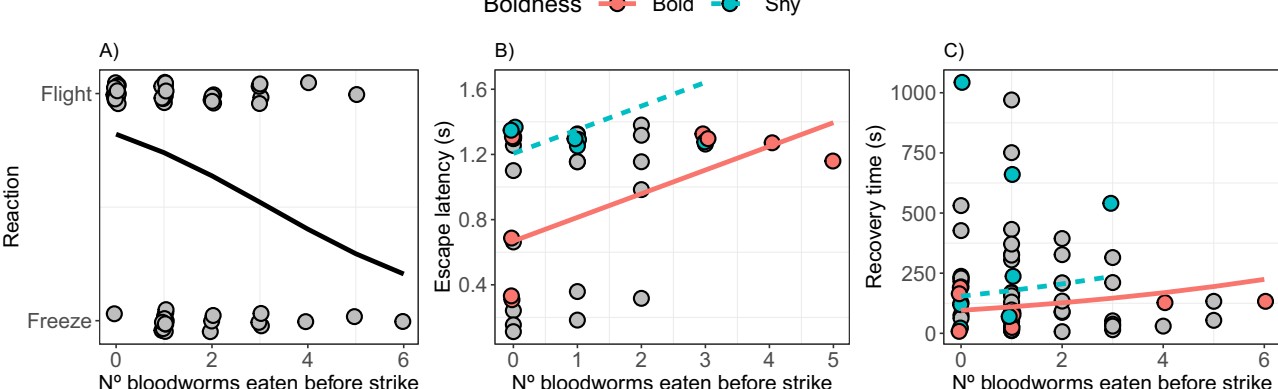

**Fig. 3 | Effect of food intake on the escape response. A** Freeze or flee reaction (Y-axis), with predicted relationship from GLMM1 (solid line), depending on the number of bloodworms eaten before the predator's attack (X-axis). Position of dots is staggered for clarity. **B** Escape latencies, with predicted relationship from GLMM2, and **C** recovery times, based on GLMM3 coefficients, depending on the number of bloodworms eaten before predator's attack. Predictions for the eight boldest (solid line) and eight shyest (dashed line) fish were obtained by using the average measurements for the eight fish with the lowest (24.8 s) or highest (1433.3 s), respectively, latency to first leave the refuge. For the recovery time, we used the predictions for the first trial day, average escape distance (32.4 cm) and average body length (3.7 cm). Dots indicate the observed values, with colouring indicating either the subset of eight boldest fish or eight shyest fish.

be perceived from the water, and placed so that its strike was directed at the centre of the food patch. The heron model was activated by a stepper motor NEMA 17 (at 20 rpm) controlled by a L298N motor driver and Arduino, and stopped just before touching the water (Supplementary Fig. 2, 3). This allowed tactile stimuli (water waves detected by the lateral line of the fish) to be minimised, and for there to be only visual cues from the stimulus. All motors were isolated from the rest of the setup to avoid any vibration cues, and we verified that no fish reacted before the predator was visible. A Logitech c920 webcam and a GoPro HERO 10 action cam were placed 80 cm above each arena, the former to track fish in real time at 30 frames per second (fps) and trigger the predator, while the latter (240 fps) was used for behavioural analysis after the trials were conducted.

## Experimental procedure

48 hours before trials, 12 fish were isolated in groups of three in square plastic mesh boxes of 19 cm depth and sides of 27 cm at the top (20 cm at the bottom), located within glass tanks identical to the holding tanks. The three fish per box were of different sizes so that they could be easily recognised by the experimenter as relatively large, medium or small fish to allow fish to be individually recognised to test for repeatability over time without the use of external tags. Each box contained a plastic tube and plastic plant to provide refuges and imitate the holding tanks. Experimental trials were carried out between 10 am and 5 pm from December 2021 to January 2022. Each fish was tested three times, 48 h apart (on Monday, Wednesday and Friday), and were fed daily, but only after trials on trial days to ensure motivation to feed in each trial. Each fish was tested in two Control tests and one 10 s test, which were tested in a random order for each fish over the three test days. During the Control tests, the predator was automatically triggered when the centre of mass of the fish was within the food patch ( ~ 1 second delay). For the 10 s condition, the predator was automatically triggered 10 ± 1 seconds after the centre of mass of the fish was detected within the food patch, allowing the fish time to consume food before the predator attacked. In case fish left the food patch before 10 seconds, the triggering of the predator was paused, and it could be triggered again if the fish came back to the food patch and remained there for 10 seconds.

Each fish ($N = 42$) was transported within a small plastic box (14 × 10 × 5 cm) with water between the holding tanks and the experimental setup ( ~ 3 m distance), and were introduced into the refuge 5 min before the start of the trial. The trial started when the door was opened and lasted 40 min. Fish were allowed to explore the arena freely and were tracked in real-time with useTracker software (https://github.com/AlexandreCampo/useTracker)[57]. Once they were within the food patch, the predator was triggered depending on the condition (Control or 10 s delay); the model was then pulled back automatically 10 seconds after the strike. If fish did visit the food patch after the strike, the predator was not triggered again, so fish could continue feeding undisturbed in order to evaluate motivation for feeding under a high level of perceived predation risk. After trials, fish were returned to their holding box, and all fish were fed *ad libitum* at the end of the day. Fish were retained in the laboratory for testing in further research after the experiment was completed.

## Ethics & inclusion

Ethical approval was granted by the University of Bristol Animal Welfare and Ethics Review Body (UIN/21/003). We have complied with all relevant ethical regulations of animal use.

## Behavioural analyses

We used useTracker software to quantify boldness and exploration from the GoPro videos at 240 fps. We used the latency to first leave the refuge as a measure of boldness (tendency to take risk)[18,21,25,58], and we used the total distance travelled (outside of the refuge) as a measure of explorative behaviour. We used the number of bloodworms eaten before the attack of the predator as a measure of food intake before escape, and as an indicator of motivation to feed. The number of bloodworms eaten after experiencing the predator's attack was used as a proxy of motivation to feed even when

predation risk is likely to be perceived as high. We also measured the latency to first reach the food patch (measured as the time elapsed from the moment fish first left the refuge and the moment fish visited the food patch).

We used DeepLabCut[59] to measure the body length of each fish and to analyse the kinematics of their escape behaviour. We first classified the fish's reaction as either freeze (did not move during the time period of the attack) or flight reaction (actively escaped from the attack) by visual observations. At the moment of the attack, we measured the distance from the predator and the angle of the predator's position relative to the orientation of the fish. If there was a flight reaction, we measured the latency to initiate the escape, the turn rate (a higher rate indicates greater agility in the response), body curvature during the escape (a lower curvature index indicates higher curvature of the body), and initial speed (indicative of the escape response's speed and acceleration) (Fig. 1). We also measured the escape distance, which refers to the distance travelled between the moment the fish first reacted to the predator and the moment the fish stopped. For all fish, we measured the recovery time as the time that elapsed between the moment the fish stopped after escaping (in the case of flight), or the end of the attack (in the case of freezing), and the moment the fish resumed their movement, as well as the distance from the refuge before resuming their activity.

## Statistics and reproducibility

We assessed repeatability of the fish's behaviour over the three trials with the rptR package[60] (with 1000 bootstraps) in R version 4.3. This package uses generalized (depending on residual distribution) linear mixed models (GLMMs) to identify consistency of individual behavioural measurements, and provides an estimation of individual repeatability ($R_C$) as the proportion of variance explained by individual identity (random factor), as well as its uncertainty[61,62]. We conducted a separate GLMM for each dependent variable: latency to first leave the refuge, total distance travelled, number of bloodworms eaten before the attack, and number of bloodworms eaten after the attack. We included trial day (1st, 2nd or 3rd), treatment (Control or 10 s delay) and body length as fixed effects, as well as individual identity as a random effect. In the case of the latency to first leave the refuge and the number of bloodworms eaten before the attack, we square-root transformed the dependent variable to achieve residual normality[63]. For the other two dependent variables, we used a GLMM with Poisson link function. The same procedure was used to analyse whether the turn rate (Gaussian distribution), the curvature index (Gaussian), the initial speed (Gaussian), the escape angle (Gaussian) and the recovery time (Poisson) were repeatable within individuals. All continuous explanatory variables were scaled (mean centred with mean = 0 and standard deviation = 1) in the models to reduce convergence issues, as well as help with numerical stability and model interpretation[64]. We performed Spearman correlation tests (with Bonferroni correction for multiple comparisons) to analyse the relationship among the latency to first leave the refuge, total distance travelled and the number of bloodworms eaten before and after the attack. Analysis of variance (ANOVA) was used to investigate whether the repeated exposure to a predator threat and learning the position of the food patch had any influence on the likelihood to engage in foraging activity.

We used GLMMs (lme4 package in R) with maximum likelihood to investigate the variables influencing escape behaviour. We focused our analysis on six different response variables. First, we conducted two GLMM analyses to investigate the non-locomotor components at the beginning of the escape response: the probability to freeze or flee, and the escape latency. In both cases, we incorporated as fixed effects boldness (latency to first leave the refuge), food intake (number of bloodworms eaten before the attack), the interaction between boldness and food intake, body length, trial day (1st, 2nd or 3rd), distance from the predator, and the angle of the attack. We chose these variables because boldness has been associated with escape latency in fish[13,20], food consumption could compromise the reaction speed or other measures of escape performance[26], and fish length could influence the perception of predation risk[65,66] or impose physical constraints during escape[67]. We added distance from the predator and angle of attack as covariates because the

position of the predator relative to the fish can determine the visual perception of the predator, as well as trial day to test if there was any habituation effect[68]. Second, we conducted three GLMM analyses for the locomotor variables during escape: turn rate, curvature index and initial speed. For all three cases, we included the same predictors as above. In addition, for the GLMM with turn rate as response variable, we included curvature as a predictor. Similarly, we included turn rate as a predictor in the GLMM with curvature index as the response variable. Regarding the initial speed, we incorporated turn rate and curvature index as predictors. Finally, we conducted a GLMM for the recovery time. In this case, we incorporated boldness and food intake (as well as their interaction), body length, trial day, escape distance, and distance from refuge before resuming activity as fixed effects. We controlled for individual identity in all models as the random effect, and all null models only included individual identity. For each response variable, we tested the effect of the explanatory variables by comparing the full model with models where each explanatory term was sequentially removed, in all possible orders. The effect of removing these variables on the second-order Akaike's information criterion (AICc) of the GLMM was used to infer which variables explained variation in the response variable. Model assumptions were checked by using model diagnostics tests (dispersion and outliers), a Kolmogorov–Smirnov test for observed vs simulated residuals, and by examining QQ-plots of the residuals generated with DHARMa package (version 0.4.6)[69] in R.

For the freeze/flight reaction and recovery time we used all individuals that triggered the predator ($N = 31$, observations = 63), although for the other escape variables we used a subset of the data including only trials where fish showed a flight reaction ($N$ fish = 25, $N$ observations = 39).

## Protocol registration
The experimental protocol was prepared in advance as part of the H2020 Marie Skłodowska-Curie IF grant, and this protocol has not been registered elsewhere.

## Reporting summary
Further information on research design is available in the Nature Portfolio Reporting Summary linked to this article.

## Data availability
All data supporting the findings of this study are available at this repository: https://doi.org/10.5281/zenodo.14783651[70].

## Code availability
The code for the data analysis is available at this repository: https://doi.org/10.5281/zenodo.14783651[70].

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

## Acknowledgements
We thank Sasha R.X. Dall for his valuable comments at the initial stages of this study. This study was supported by a H2020 Marie Skłodowska-Curie IF grant (DynFish – 891915) to I. Planas-Sitjà and C. C. Ioannou, and a Natural Environment Research Council grant number NE/P012639/1 to C. C. Ioannou.

## Author contributions
Conceptualization, Methodology, Funding acquisition, Resources: I. Planas-Sitjà and C. C. Ioannou. Investigation, Data curation, Formal analysis, Software, Writing - original draft: I. Planas-Sitjà. Writing - review & editing: C. C. Ioannou.

## Competing interests
All authors declare no competing interests.
