## [Transparent Peer Review file · Communications Biology]

State-behaviour feedbacks between boldness and food intake shape escape responses in fish (*Gasterosteus aculeatus*)

Corresponding Author: Dr Isaac Planas-Sitjà

Version 0:

Reviewer comments:

Reviewer #1

(Remarks to the Author)

This manuscript aims to investigate the effect of food consumption on escape response parameters (freeze/flee, latency to escape, turn rate and latency to resume foraging) and the interaction with “boldness”. It is predicted that consuming food will slow down escape responses, particularly in bolder fish which are predicted to prioritise food consumption. The key results indicate that fish switch from freezing to fleeing if they consume more food and have a higher latency to escape after consuming more food. Bolder fish also have a lower escape latency (react quicker) than shier fish.

Overall, this is an interesting paper that builds on the idea of escape being strategic and economic and linked to life-history trade-offs. I like that the authors have considered a range of escape response parameters i.e. as well as latency to escape, they measured turn rate, curvature index and type of response (freeze/flee).

However, there are a few key limitations with this manuscript.

1. The a-priori predictions could be articulated and evidenced more clearly. It is not clear where the evidence base is for predicting that bolder fish will react faster than shyer fish. One paper is cited to suggest bolder fish react faster due to higher metabolisms, but it is suggested “it has rarely been considered that bolder individuals may be less likely to respond, or respond more slowly, to a predatory threat because they allocate a greater proportion of their attention to foraging rather than detecting predators”. Paper 49 in your ref list (Jones and Godin, 2010), specifically tests the hypothesis that personality and escape response delay may be linked. While not a direct repeat of this manuscript (the personality measures were different), it clearly conveys the idea of divided attention and bolder “riskier” personalities potentially prioritising food leading to slower escape responses. It also suggests the counterargument of bolder personalities responding faster due to compensation for increased risk taking. Overall, I think you need to better map the literature to your linkage between boldness and escape response delays and I’m not convinced you can a-priori suggest from the current evidence base that bolder fish will react faster (this feels like “HARKing”). It would be better to set out hypotheses for both a positive and negative relationship since you essentially start that narrative on line 32 before settling on bolder fish reacting faster on line 59 (without any reasoning why)

2. The analysis of personality measurements are hard to follow. It is clear what behavioural tests you have used to quantify personality and it is good that you clearly present how repeatable these traits are. However, it is hard to follow at times how your personality measurements are treated in your results. For example, in the methods, it is made clear that your measure of “boldness” is latency to leave the refuge, treated as a fixed continuous effect. I have no problem with this, since it was repeatable, is consistent with the literature, and it’s the only measure you really have that is independent of food measures. However, in your figures, you are instead categorising fish as either “bold” or “shy” and the categorisation is only in the legend (not in main methods) and in-text you simply refer to “bold fish reacted faster than shy fish”. I think it would be clearer to refer to the actual method of measuring boldness (latency to leave refuge) in the reporting of the results, to make it clearer how boldness was measured without having to dig through the manuscript methods (especially with this style of paper where the methods follow the results). I’m also not sure what the PCA really adds to the analysis – this is perhaps partly why it was confusing understanding how boldness was measured in the results, since typically you do a PCA to reduce many different measurements of behaviours down to a few axes. So I don’t really understand why you would do a PCA and then

not use those PC1 and PC2 axes in subsequent analyses and I am also not sure you have enough different behavioural measures to warrant a PCA, particularly since many measures also link to food consumption which is a main effect you are trying to independently link to personality so it seems rather circular and non-independent to include it in your measures of personality. You have basically just measured boldness via latency to leave the refuge and so I'd just stick with that and make the manuscript clearer and simpler.

3. The measurement of boldness is not independent of the experiment designed to link boldness to food intake and escape. For example, while it is good that there are repeated measures of the latency to leave the refuge (key for quantifying personality), this personality measure was conducted as part of the predator attack experiment, which means the 2nd and 3rd latency to leave the refuge measurements were taken after attacks. So it's not really the same test repeated 3x since knowledge of risk is changing across the 3 trials (in trials 2 and 3 the fish may "know" to expect a heron attack). Ideally, this experiment could have been designed to quantify personality first, and then do the foraging and predator attack experiments second, rather than interweaving the two. I'm not sure this is a massive problem but it needs more acknowledgement. In some ways it would be interesting to see the individual fish latencies plotted for trials 1,2,3 (i.e. a reaction-norm type plot). This would show the inter-individual variance but also how it interacts with intra-individual variance (plasticity with increasing risk e.g. see box 2 of Dingemanse et al (2009) <https://doi.org/10.1016/j.tree.2009.07.013>).

4. This paper does not conform to the ARRIVE guidelines for reporting of animal use which are "strongly encouraged" by this journal. I would encourage the authors to look at the 2020 (ARRIVE 2.0) paper on this. It is a simple list to make sure key aspects of animal use are reported (it is not enough to simply state that you had ethical approval). A noticeable omission in this manuscript is a lack of clear reporting of the number of animals used. While it is clear that 42 fish were put in the experimental set-up it is not clear how many fish were initially captured and how fish were selected for the experiment and if any fish died over the experimental time period (including the two months prior). (For reference, I have post-hoc since ARRIVE gone back through my old papers and found I did not meet ARRIVE in several areas, so we all have work to do in this area – the point of ARRIVE is to try to make reporting quality more transparent and consistent across journals/papers, as without prompt lists it has been demonstrated that many scientists forget to report important information regarding animal use).

5. This is probably old-fashioned of me, but I found the results hard to interpret without p-values. I understand that there is debate about the use of p-values, particularly for mixed models. Hence, some authors may choose not to report them. However, I find it hard to just interpret parameter estimates reported in-text alongside the authors judgement on which effects are more important or having to dig through supplementary material. This feels rather descriptive and hidden. If not using p-values, perhaps an alternative to help guide the readers understanding of what variables are more significant, would be to use an effect size plot (which would could replace the current PCA plot) so at a quick glance it can be seen which variables have stronger effects and which confidence intervals cross zero etc. Alternatively, if you absolutely must determine in the way you have chosen, evidence it with some papers that suggest this is how the field is shifting towards statistical reporting, as you have done effectively for using Q-Q plots to check residuals (which I do agree with, and think it's nice you've reported model checking).

More specific line-by-line minor points.

Line 112. I found the result that turn rates were faster when at a further distance from the predator very interesting – this is consistent with literature on pursuit-deterrent signals – e.g. behaviours like stotting tend to occur at distances further from the predator (when it's perhaps less risky to "jump about"). It also links a bit to the literature on protean escape – essentially a more erratic escape path can have benefits (harder to predict), but perhaps at a cost of speed of escape. So your results indicate a more protean strategy when not requiring last-ditch fast escape? If you still have this lab set-up it would be interesting to explore turning rates a bit more in another study even if they haven't linked in with the personality narrative of this manuscript. It's also interesting that turning rate is repeatable – this suggests fish are consistent in how "erratic" their escape pattern is?

Line 167. I think you could also link to response type here, since fish switched more to freezing after eating more. This again indicates that ingested food leads to a less "active" escape strategy?

Line 182. I think it's nice to compare across taxa but the Arctic Fox example here seemed a bit tangential to your study.

Line 184. You mention reaction norms here, and I've mentioned them above. As part of your experimental design do you have reaction norms? Since you have control trials vs. 10s trials (differences in food eaten) and trials 1, 2, 3 (differences in exposure to predator model). So you could somewhat look at the effect of hunger on escape and effect of perceived predation risk on escape at an individual level? Did you try any reaction norm plots?

Line 195. "The effects of non-linearities, such as the relationship between mortality risk and risky behaviour, should be considered when investigating the maintenance of personality variance within populations". At the end of this paragraph you start talking about the link between personality traits and fitness (via mortality) and maintenance of variation in the population. These are quite broad, general points rather than really digging into your experimental findings, and as a take-home message I found this sentence rather inaccessible (I had to read it several times and think about it) and a bit generic. The word count is quite tight, so I would focus the narrative as much as possible on what is novel or interesting about your study.

Line 303. Explain further why and how variables were scaled (i.e. justify more).

Line 37, 38, 40. Some references missing information e.g. page numbers/articleID or volume number. Otherwise seems to follow journal guidelines. The refs are appropriate and contain most key papers in the field. Only notable omission is perhaps Ydenberg and Dill who were important in putting forward the idea of strategic escape response delays, and could perhaps the idea of economic escape could be woven into the opening paragraph (although I appreciate the word count is tight!).

Reviewer #2

(Remarks to the Author)

I really enjoyed reading this manuscript! The question of how different tactics are maintained within a population is very interesting and should be of considerable interest to the journal's readership. I have only a few minor suggestions to improve the manuscript.

Line 11: wouldn't 'shy' vs. 'bold' be technically considered tactics within the same 'strategy'?

Line 20 (and elsewhere): does Szopa-Comley & Ioannou (2022) provide contradictory findings? It would be useful to the reader to have this information included in the text in order to fully understand the argument being made (there are a few instances of this in the introduction).

Line 30: while it is likely that growth-mortality trade-offs can maintain variation, it is also the case that predation risk might shape personality (high risk leading to bolder phenotypes; Culum Brown's work for example). If the trade-offs mentioned mediate individual responses, wouldn't they in turn shape perceived risk? I guess I'm wondering what the role of individual experience might be in maintaining personality types within a population.

Line 48-50: again, I'm wondering what the experiential role might be in shaping these trade-offs?

Figure 2 (and related text): it would greatly benefit the reader if both PC1 and PC2 were better described in the text. For example, high values of PC1 would reflect shorter escape latencies + higher exploration + more foraging? Also, PC2 reflects what?

Line 193-195: Perhaps you could expand on this point. There are a number of examples of behavioural differences in the predator avoidance responses in shy vs. bold, but there are also examples of differences in underlying cognitive mechanisms (shyer individuals learn and retain information concerning risk differently than do bolder individuals). Such cognitive differences would shape the trade-offs discussed in the current paper.

Line 207: more information regarding the collection site is needed. For example, how many shoals were collected (one large or many small); was it a high predation risk site; characterize the microhabitat (complex vs. simple). Such biotic factors are known to shape personality.

Line 223: would individuals have the opportunity to learn where the foraging patch is located? Would this impact the behaviour recorded?

Again, I read this paper with great interest and I look forward to seeing it in print!

Author Rebuttal letter:

Reviewers' comments:

Reviewer #1 (Remarks to the Author):

This manuscript aims to investigate the effect of food consumption on escape response parameters (freeze/flee, latency to escape, turn rate and latency to resume foraging) and the interaction with "boldness". It is predicted that consuming food will slow down escape responses, particularly in bolder fish which are predicted to prioritise food consumption. The key results indicate that fish switch from freezing to fleeing if they consume more food and have a higher latency to escape after consuming more food. Bolder fish also have a lower escape latency (react quicker) than shier fish.

Overall, this is an interesting paper that builds on the idea of escape being strategic and economic and linked to life-history trade-offs. I like that the authors have considered a range of escape response parameters i.e. as well as latency to escape, they measured turn rate, curvature index and type of response (freeze/flee).

However, there are a few key limitations with this manuscript.

1. The a-priori predictions could be articulated and evidenced more clearly. It is not clear where the evidence base is for predicting that bolder fish will react faster than shyer fish. One paper is cited to suggest bolder fish react faster due to higher metabolisms, but it is suggested "it has rarely been considered that bolder individuals may be less likely to respond, or respond more slowly, to a predatory threat because they allocate a greater proportion of their attention to foraging rather than detecting predators". Paper 49 in your ref list (Jones and Godin, 2010), specifically tests the hypothesis that personality and escape response delay may be linked. While not a direct repeat of this manuscript (the personality measures were different), it clearly conveys the idea of divided attention and bolder "riskier" personalities potentially prioritising food leading to slower escape responses. It also suggests the counterargument of bolder personalities responding faster due to compensation for increased risk taking. Overall, I think you need to better map the literature to your linkage between boldness and escape response delays and I'm not convinced you can a-priori suggest from the current evidence base that bolder fish will react faster (this feels like "HARKing"). It would be better to set out hypotheses for both a positive and negative relationship since you essentially start that narrative on line 32 before settling on bolder fish reacting faster on line 59 (without any reasoning why)

**The line numbers provided correspond to the tracked changes version

Following the advice of reviewer 1, we have modified the introduction to better map to previous literature, and added more information regarding our hypothesis (L38-57; L73-78).

Regarding the work of Jones and Godin 2010, they indeed show that bold individuals (more willing to leave a companion and explore a novel object) react slower to a predator in the context of foraging. In a different context (i.e., satiated and in a novel environment without food to forage on), it is plausible that bold fish would have reacted faster. In addition, it should be noted that prey were attacked during foraging (after ~30s), but without measuring the amount of food ingested. We cannot exclude the effect of food intake in their experiment, as if more exploratory fish consumed more food than less exploratory fish, that could also explain differences in escape reaction. Finally, just for clarification, while those authors suggest that their results can be due to attention bias, they do not rule out the possibility that differences in neurological or cognitive capacities could explain their results too. As reviewer 1 mentions, there are two key differences between their study and ours that makes the comparison difficult:

1. Jones and Godin 2010 did not measure repeatability of escape responses.
2. They did not analyse the effects of food intake.

2. The analysis of personality measurements are hard to follow. It is clear what behavioural tests you have used to quantify personality and it is good that you clearly present how repeatable these traits are. However, it is hard to follow at times how your personality measurements are treated in your results. For example, in the methods, it is made clear that your measure of "boldness" is latency to leave the refuge, treated as a fixed continuous effect. I have no problem with this, since it was repeatable, is consistent with the literature, and it's the only measure you really have that is independent of food measures. However, in your figures, you are instead categorising fish as either "bold" or "shy" and the categorisation is only in the legend (not in main methods) and in-text you simply refer to "bold fish reacted faster than shy fish". I think it would be clearer to refer to the actual method of measuring boldness (latency to leave refuge) in the reporting of the results, to make it clearer how boldness was measured without having to dig through the manuscript methods (especially with this style of paper where the methods follow the results). I'm also not sure what the PCA really adds to the analysis – this is perhaps partly why it was confusing understanding how boldness was measured in the results, since typically you do a PCA to reduce many different measurements of behaviours down to a few axes. So I don't really understand why you would do a PCA and then not use those PC1 and PC2 axes in subsequent analyses and I am also not sure you have enough different behavioural measures to warrant a PCA, particularly since many measures also link to food consumption which is a main effect you are trying to independently link to personality so it seems rather circular and non-independent to include it in your measures of personality. You have basically just measured boldness via latency to leave the refuge and so I'd just stick with that and make the manuscript clearer and simpler.

We understand that the use of the principal components might have been misleading. The PCA was merely to show the relationship among response variables, and boldness was meant to refer to the latency to first leave the refuge in the rest of the manuscript. We have modified the results section to make this clearer: we have removed the PCA (L128), added a new figure 2 (L130) to reflect the relationships among the four main variables explored, and modified the text at the beginning of the results section to clarify the meaning of boldness (L96-100). We kept the categorisation within figures merely for visual purposes (as they are based on the latency to leave the refuge too), and we have modified the figure legend to reflect this (L160).

3. The measurement of boldness is not independent of the experiment designed to link boldness to food intake and escape. For example, while it is good that there are repeated measures of the latency to leave the refuge (key for quantifying personality), this personality measure was conducted as part of the

predator attack experiment, which means the 2nd and 3rd latency to leave the refuge measurements were taken after attacks. So it's not really the same test repeated 3x since knowledge of risk is changing across the 3 trials (in trials 2 and 3 the fish may "know" to expect a heron attack). Ideally, this experiment could have been designed to quantify personality first, and then do the foraging and predator attack experiments second, rather than interweaving the two. I'm not sure this is a massive problem but it needs more acknowledgement. In some ways it would be interesting to see the individual fish latencies plotted for trials 1,2,3 (i.e. a reaction-norm type plot). This would show the inter-individual variance but also how it interacts with intra-individual variance (plasticity with increasing risk e.g. see box 2 of Dingemans et al (2009) <https://doi.org/10.1016/j.tree.2009.07.013>).

Regarding the repeatable measures, while it is true that there are other ways of testing them, we do not believe this is a major issue, for this specific study at least, as the potential effect of "day" is included in all GLMMs performed. We could have conducted repeated tests without the predator, but then we increased the chances of having the influence of processes such as learning, habituation or sensitization, which would be difficult to control for. As in all personality studies, we had to make a choice to balance the extent of repeating testing, and we chose to decrease the potential effects of habituation to the setup. To better see the effects of predator exposure, as suggested, we have added reaction norm plots as supplementary figure S1, and added results demonstrating no effect of repeated testing on a number of variables (L86-95).

4. This paper does not conform to the ARRIVE guidelines for reporting of animal use which are "strongly encouraged" by this journal. I would encourage the authors to look at the 2020 (ARRIVE 2.0) paper on this. It is a simple list to make sure key aspects of animal use are reported (it is not enough to simply state that you had ethical approval). A noticeable omission in this manuscript is a lack of clear reporting of the number of animals used. While it is clear that 42 fish were put in the experimental set-up it is not clear how many fish were initially captured and how fish were selected for the experiment and if any fish died over the experimental time period (including the two months prior). (For reference, I have post-hoc since ARRIVE gone back through my old papers and found I did not meet ARRIVE in several areas, so we all have work to do in this area – the point of ARRIVE is to try to make reporting quality more transparent and consistent across journals/papers, as without prompt lists it has been demonstrated that many scientists forget to report important information regarding animal use).

Thanks for the advice. We modified the manuscript to adhere to the ARRIVE guidelines as much as possible (see L260-275).

5. This is probably old-fashioned of me, but I found the results hard to interpret without p-values. I understand that there is debate about the use of p-values, particularly for mixed models. Hence, some authors may choose not to report them. However, I find it hard to just interpret parameter estimates reported in-text alongside the authors judgement on which effects are more important or having to dig through supplementary material. This feels rather descriptive and hidden. If not using p-values, perhaps an alternative to help guide the readers understanding of what variables are more significant, would be to use an effect size plot (which would could replace the current PCA plot) so at a quick glance it can be seen which variables have stronger effects and which confidence intervals cross zero etc. Alternatively, if you absolutely must determine in the way you have chosen, evidence it with some papers that suggest this is how the field is shifting towards statistical reporting, as you have done effectively for using Q-Q plots to check residuals (which I do agree with, and think it's nice you've reported model checking). We have added p-values in the results when needed, and a new figure 2. We have also removed the PCA (L128). We hope these changes will make the results more clear.

More specific line-by-line minor points.

Line 112. I found the result that turn rates were faster when at a further distance from the predator very interesting – this is consistent with literature on pursuit-deterrent signals – e.g. behaviours like stotting tend to occur at distances further from the predator (when it's perhaps less risky to "jump about"). It also links a bit to the literature on protean escape – essentially a more erratic escape path can have benefits (harder to predict), but perhaps at a cost of speed of escape. So your results indicate a more protean strategy when not requiring last-ditch fast escape? If you still have this lab set-up it would be interesting to explore turning rates a bit more in another study even if they haven't linked in with the personality narrative of this manuscript. It's also interesting that turning rate is repeatable – this suggests fish are consistent in how "erratic" their escape pattern is?

Thanks for the suggestion. It would be interesting indeed to investigate the "erratic" of such trait. Regarding the second question ("this suggests fish are consistent in how 'erratic' their escape pattern is?"), it is difficult to provide a definitive answer with the current data. We believe that the repeatability of the turning rate is probably most indicative of the physical condition or the energy invested in the escape movement. For instance, fish can have same/similar turn rate during escapes, but different escape directions/angles.

Line 167. I think you could also link to response type here, since fish switched more to freezing after eating more. This again indicates that ingested food leads to a less “active” escape strategy?
We have added this information (L208).

Line 182. I think it's nice to compare across taxa but the Arctic Fox example here seemed a bit tangential to your study.
We have modified this section (see L 224-229).

Line 184. You mention reaction norms here, and I've mentioned them above. As part of your experimental design do you have reaction norms? Since you have control trials vs. 10s trials (differences in food eaten) and trials 1, 2, 3 (differences in exposure to predator model). So you could somewhat look at the effect of hunger on escape and effect of perceived predation risk on escape at an individual level? Did you try any reaction norm plots?

We did do some preliminary plots to examine this. However, there is no real gradient between the control and 10s trials (i.e., 10s just allows more time to eat than the control), and because all fish were fed equally the day before, it would be difficult to estimate their hunger level from our data. We have now added reaction norm plots over repeated trials instead (fig. S1). However, the interpretation of the plots is quite difficult as it has confounding factors. Perception of risk could decrease due to habituation to the setup; or increase due to sensitization to the predator if they triggered the attack; or decrease due to habituation to the predator stimulus. In any case, we did not see any trends in our data, except that fish tend to show higher plasticity regarding distance travelled compared to other measurements (supplementary figure S1; L93; L189).

Line 195. “The effects of non-linearities, such as the relationship between mortality risk and risky behaviour, should be considered when investigating the maintenance of personality variance within populations”. At the end of this paragraph you start talking about the link between personality traits and fitness (via mortality) and maintenance of variation in the population. These are quite broad, general points rather than really digging into your experimental findings, and as a take-home message I found this sentence rather inaccessible (I had to read it several times and think about it) and a bit generic. The word count is quite tight, so I would focus the narrative as much as possible on what is novel or interesting about your study.

We have modified this paragraph (L 240-250).

Line 303. Explain further why and how variables were scaled (i.e. justify more).

The explanatory variables have different orders of magnitude, and because we use quite complex models, scaling variables help reduce convergence issues and with the interpretation of the returned values. We have added this information at L 367.

Line 37, 38, 40. Some references missing information e.g. page numbers/articleID or volume number. Otherwise seems to follow journal guidelines. The refs are appropriate and contain most key papers in the field. Only notable omission is perhaps Ydenberg and Dill who were important in putting forward the idea of strategic escape response delays, and could perhaps the idea of economic escape could be woven into the opening paragraph (although I appreciate the word count is tight!).

Thank you for the comments on the references. We have revised the reference list. We are familiar with Ydenberg and Dill's work; sadly, we added several references to accommodate comments from reviewer 1 and 2, and the list is already longer than the recommended length of the journal.

Reviewer #2 (Remarks to the Author):

I really enjoyed reading this manuscript! The question of how different tactics are maintained within a population is very interesting and should be of considerable interest to the journal's readership. I have only a few minor suggestions to improve the manuscript.

Line 11: wouldn't 'shy' vs. 'bold' be technically considered tactics within the same 'strategy'?

**The line numbers provided correspond to the tracked changes version

Indeed, in this context tactics would be a more appropriate term; we have made this change (L12).

Line 20 (and elsewhere): does Szopa-Comley & Ioannou (2022) provide contradictory findings? It would be useful to the reader to have this information included in the text in order to fully understand the argument being made (there are a few instances of this in the introduction).

Szopa-Comley and Ioannou (2022) used artificial prey with predictable or unpredictable escape responses to real predator, and their results were contradictory in the sense that they did not find any

effect of prey predictability in capture rates. However, because this reference was only cited here, and was not really relevant for the rest of the paragraph, we decided to remove it (the list of references is quite long already).

Line 30: while it is likely that growth-mortality trade-offs can maintain variation, it is also the case that predation risk might shape personality (high risk leading to bolder phenotypes; Culum Brown's work for example). If the trade-offs mentioned mediate individual responses, wouldn't they in turn shape perceived risk? I guess I'm wondering what the role of individual experience might be in maintaining personality types within a population. Indeed, we intended to include predation risk within 'mortality'. We have modified this paragraph to include this idea in the new version (L33-36). It is a good question, although we are afraid that the answer to this question is out of the scope of this paper.

Line 48-50: again, I'm wondering what the experiential role might be in shaping these trade-offs? We have added reaction norm plots in supplementary figure S1, and performed ANOVAs to provide some information regarding the role of risk perception and learning the position of the food patch over the repeated trials. We did not see any trend that could suggest a change in behaviour due to experience of repeated trials (L86-95 and figure S1).

Figure 2 (and related text): it would greatly benefit the reader if both PC1 and PC2 were better described in the text. For example, high values of PC1 would reflect shorter escape latencies + higher exploration + more foraging? Also, PC2 reflects what?

Following comments from reviewer 1, we have removed this analysis from the manuscript, and instead we added a new figure 2 (L130). Reviewer 2 was correct: PC1 reflected differences in food intake, distance travelled and escape latency, while PC2 mainly reflected differences among fish regarding distance travelled; some fish travelled long distances and did not eat (mainly moving along the borders of the arena), while others went to the food patch, ate, and went back to the refuge. We have also added this information in the new version (Figure S1, L93, L189)

Line 193-195: Perhaps you could expand on this point. There are a number of examples of behavioural differences in the predator avoidance responses in shy vs. bold, but there are also examples of differences in underlying cognitive mechanisms (shyer individuals learn and retain information concerning risk differently than do bolder individuals). Such cognitive differences would shape the trade-offs discussed in the current paper.

We have added text regarding the underlying cognitive mechanisms at L240-250.

Line 207: more information regarding the collection site is needed. For example, how many shoals were collected (one large or many small); was it a high predation risk site; characterize the microhabitat (complex vs. simple). Such biotic factors are known to shape personality. We have included more information about the collection site (L260-275).

Line 223: would individuals have the opportunity to learn where the foraging patch is located? Would this impact the behaviour recorded?

That is a good question. The patch is always at the same place, so as long as they have fed during previous trials, there is a possibility for fish to learn the position of the patch. However, we did not see any effect of testing day in any model (except two cases with a weak effect), and we did not observe any trend in the latency to reach the food patch over repeated trials (L86-95 and figure S1).

Again, I read this paper with great interest and I look forward to seeing it in print!
Thank you very much.
